# RRVF: Visual Reinforcement Learning with Reasoning, Rendering, and Visual Feedback

## Abstract

Multimodal Large Language Models (MLLMs) exhibit impressive performance across various visual tasks. Subsequent investigations into enhancing their visual reasoning abilities have significantly expanded their performance envelope. However, a critical bottleneck in the advancement of MLLMs toward deep visual reasoning is their heavy reliance on curated image-text supervision. To solve this problem, we introduce a novel framework, *"Reasoning-Rendering-Visual-Feedback" (RRVF)*, that enables MLLMs to learn complex visual reasoning from only raw images. This framework builds on the *"Asymmetry of Verification"* principle, *i.e.*, verifying the rendered output against the source image is substantially easier than performing deep visual reasoning to generate a faithful, structured representation such as code. We demonstrate that this relative ease provides an ideal reward signal for optimization via Reinforcement Learning (RL), thereby reducing reliance on image-text supervision. RRVF implements a closed-loop iterative process encompassing reasoning, rendering, and visual feedback components, enabling the model to perform complex reasoning, including self-correction through multi-turn interactions. This process is optimized end-to-end using the GRPO algorithm. Extensive evaluations are conducted on image-to-code generation across two diverse domains: data charts and web interfaces. The RRVF-trained model not only outperforms existing similarly sized open-source MLLMs and supervised fine-tuning baselines but also exhibits superior generalization. Notably, the model outperforms the more advanced MLLM used to generate visual feedback during training. The code will be made publicly available.

## 1 Introduction

Improving complex reasoning in Large Language Models (LLMs) represents a fundamental challenge in AI research and a critical milestone on the path to Artificial General Intelligence (AGI) (OpenAI, 2024b; DeepSeek-AI et al., 2025; Hu et al., 2025; Team et al., 2025b). Recent works have attempted to replicate this success in Multimodal Large Language Models (MLLMs) for addressing complex visual tasks, yet this paradigm introduces several challenges (Huang et al., 2025; Shen et al., 2025a).

Specifically, for MLLMs, effective reasoning necessitates capabilities that extend beyond elementary visual perception to achieve sophisticated visual comprehension and inference. This paradigm requires models to not only perform object recognition but also to analyze underlying visual semantics, spatial-geometric relationships, and complex inter-object dependencies (Su et al., 2025c; Luo et al., 2024). While

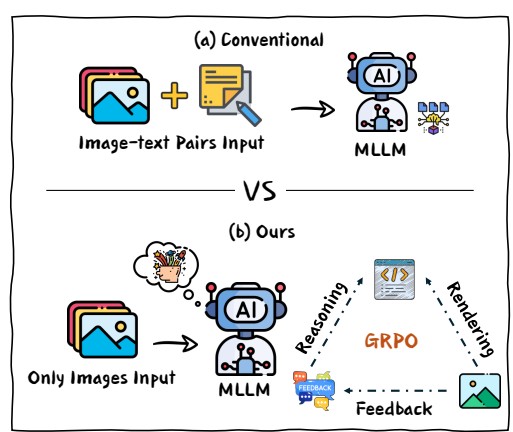

Figure 1: Comparison of training paradigms. Our RRVF (bottom) trains solely on raw images via a closed-loop reasoning-rendering-visual-feedback process under GRPO.

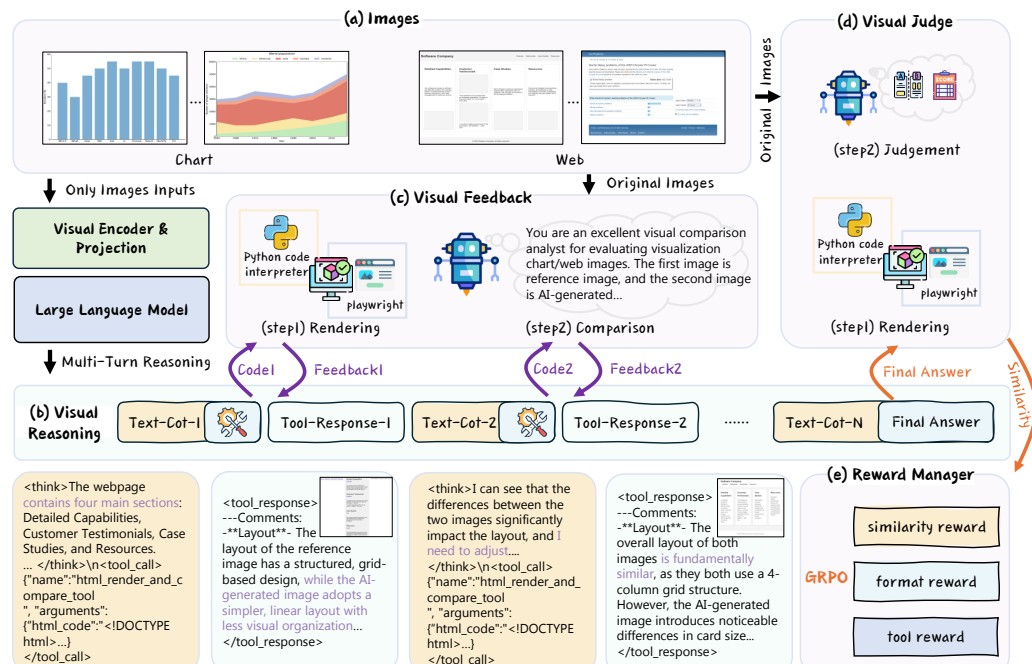

Figure 2: RRVF is a training framework that improves visual reasoning ability using only image inputs. Given an input image (a), the multimodal language model performs iterative reasoning (b) by generating rendering code. The code is executed by external tools, and the output is compared with the original image by a visual judge (d). The comparison results are converted into structured feedback (c), which guides the next round of code generation. The reward manager (e) combines visual similarity, format validity, and tool usage efficiency. These signals are used to optimize the model through reinforcement learning, implemented with the GRPO algorithm.

accurate visual perception serves as a fundamental prerequisite for such reasoning capabilities, it inherently depends on robust reasoning mechanisms, creating a challenging bidirectional dependency where perception and reasoning are mutually reinforcing.

Recently, the pioneering work, like OpenAI's "Think with Images" (OpenAI, 2025b; Su et al., 2025a), has shown how external tools like OCR and zoom can enable models to interact proactively with visual content. Subsequent approaches expand beyond limited tool sets by generating executable code for visual operations through Reinforcement Learning (RL) training (Sutton et al., 1998), as seen in VRAG-RL (Wang et al., 2025b; Liu et al., 2025b). This demonstrates that incorporating external tools and reusing visual representations in MLLM reasoning can effectively enhance model performance.

Inspired by this, we propose an MLLM RL framework called "Reasoning-Rendering-Visual-Feedback" (RRVF) to endow MLLMs with enhanced visual comprehension and reasoning capabilities. As shown in Figure 1, this framework enhances visual understanding through a cycle of reasoning, rendering, and visual feedback, relying solely on images as training data. The core design is motivated by the "Asymmetry of Verification" principle (Wei, 2025), *i.e.*, for many tasks, a proposed solution is substantially easier to verify than to generate. Specifically, tasks exhibiting this property are particularly amenable to RL training, as the efficient verification process can be directly formulated as a reward signal. Remarkably, we observe that for `image-to-code` tasks, *e.g.*, converting the chart image to the executable code, verifying the visual similarity between rendered outputs and source images proves substantially more tractable than generating rendering code from scratch.

Within the RRVF framework, the MLLM reasons about an input image to generate rendering code, which is then executed by a tool. Based on the visual discrepancy between the rendered and original images, the MLLM iteratively refines its code across multiple turns. To optimize this closed-loop

refinement process, the system is modeled as a reinforcement learning task driven by the GRPO algorithm Shao et al. (2024b). The learning is guided by a reward signal that incorporates the final visual similarity score. This mechanism compels the model to understand the image's generative logic from raw pixels, leading to semantically correct code. The training process requires no text-based ground truth. This naturally resolves the "semantic equivalence" issue, as any program that produces the correct visual output is rewarded.

Moreover, the GRPO optimization is guided by a hybrid reward function. (1) The primary component is a visual similarity reward that quantifies the fidelity between the rendered image and the original image by integrating signals from established vision-language models. (2) This is supplemented by a format correctness reward to ensure the generation of syntactically correct and executable code. (3) An adaptive tool-use reward is introduced to balance exploration and convergence. This hybrid reward function effectively enables the model to learn underlying generative logic directly from pixel-level visual representations.

To demonstrate the effectiveness of the proposed RRVF framework, we evaluate our method in two distinct domains, data charts and web interfaces, while *using only raw pixel-based images as input*. Extensive experiments show that the RRVF framework enables MLLMs to effectively learn the underlying generative logic for both domains, capturing structures that range from the strict syntax of programming toolkits to the flexible layouts of web pages.

Our main contributions are as follows:

- The exploration of the "Asymmetry of Verification" principle for MLLM training in complex visual reasoning tasks. We establish that verifying the visual correctness of a rendered output against a source image is a significantly more tractable problem than generating the initial code. This asymmetry provides an effective learning signal that circumvents the need for textual supervision.

- A novel framework, termed Reasoning-Rendering-Visual-Feedback (RRVF), is proposed. It implements a closed-loop iterative process that enables model self-correction through multi-turn interactions and tool invocation, and this pipeline is optimized in an end-to-end manner using the GRPO algorithm.

- Extensive evaluation of the RRVF framework across two structurally diverse domains (i.e., data charts and web interfaces) demonstrates that the proposed approach successfully learns underlying generative logic directly from pixel-level representations. The model trained with RRVF significantly outperforms state-of-the-art open-source MLLMs of comparable scale as well as supervised fine-tuning approaches on image-code pairs, exhibiting superior generalization on unseen datasets. Notably, it also surpasses the larger model used during training to provide visual feedback and judgment. This establishes a promising paradigm for developing more capable and robust MLLMs.

## 2 RELATED WORKS

### 2.1 MULTIMODAL LARGE MODELS

Multimodal Large Language Models (MLLMs) are rapidly evolving (Liu et al., 2023; Bai et al., 2025; Zhu et al., 2025; Anthropic, 2025; OpenAI, 2025a; Google, 2025), demonstrating remarkable capabilities in visual understanding and cross-modal interaction. At the forefront are powerful proprietary models such as OpenAI o3 (OpenAI, 2025a) and Gemini-2.5 (Google, 2025), which employ a unified Transformer architecture for end-to-end processing of diverse inputs, including images, text, and long videos. Concurrently, the open-source community is making significant strides (Team et al., 2025c; Guo et al., 2025; Team et al., 2025a). For instance, InternVL3 (Zhu et al., 2025) has achieved state-of-the-art (SOTA) performance on numerous multimodal benchmarks through its robust visual encoder and refined alignment mechanism. Similarly, models like Qwen2.5-VL (Bai et al., 2025) have shown strong capabilities in multi-language support and specialized tasks like optical character recognition (OCR) and document understanding. These collective efforts, from both proprietary and open-source domains, have significantly advanced the field and promoted the democratization of related technologies (Chu et al., 2025; Liu et al., 2025a). Despite these advances in visual representation and long-context processing, current MLLMs still have limitations in deep

---

**Algorithm 1** Iterative Visual Reasoning

---

**Require:** MLLM $\mathcal{M}$, Single Image $I$, Max Turns $T_{max}$
**Ensure:** Final rendering code $C_{final}$
1: $H \leftarrow \text{InitialPrompt}(I)$
2: **for** $t = 1$ to $T_{max}$ **do**
3:     $R_{model} \leftarrow \mathcal{M}.\text{Generate}(H)$                                      ▷ Generate response
4:     $H \leftarrow H \oplus R_{model}$
5:     **if** $R_{model}$ contains `<answer>` **then**
6:         $C_{final} \leftarrow \text{Extract}(R_{model}, \text{<answer>})$
7:         **break**                                    ▷ Final turn: task is complete
8:     **else**
9:         $C_{tool} \leftarrow \text{Extract}(R_{model}, \text{<tool\_call>})$
10:        $F \leftarrow \text{CallTool}(C_{tool})$                          ▷ Invoke the external tool
11:        $H \leftarrow H \oplus F$                          ▷ Append visual feedback to history
12:     **end if**
13: **end for**
14: **return** $C_{final}$

---

visual reasoning and do not fully utilize visual information (Su et al., 2025c; Wang et al., 2025c; Mao et al., 2025).

## 2.2 VISUAL REASONING

Visual reasoning, as a core capability in multimodal understanding, has garnered significant attention from the academic community in recent years Su et al. (2025c). Efforts to bridge the semantic gap between continuous visual information and discrete language have evolved through several stages: from encoding images into static visual features for Large Language Models (LLMs), to leveraging text-based Chain-of-Thought (CoT) for reasoning Zhang et al. (2023); Wang et al. (2025c), which explicitly breaks down complex visual problems into textual intermediate steps (Shao et al., 2024a), and more recently, to accomplishing tasks by invoking external tools Zheng et al. (2025); Xu et al. (2025); Wu et al. (2024b); Fu et al. (2025). This paradigm, where models learn to call APIs, was pioneered in language-only contexts by works like Toolformer (Schick et al., 2023) and extended to multimodal scenarios (Wu et al., 2023). Building upon this, some models act as advanced planners Qi et al. (2024); Su et al. (2025b), enriching visual understanding at a finer-grained level by iteratively invoking tools like "zoom in" and "crop" as reasoning steps. To further enhance the model's autonomous cognitive abilities, subsequent research has transcended the limitations of predefined toolsets by generating executable visual code (Beltramelli, 2018; Zhao et al., 2025; Wang et al., 2025a; Shen et al., 2025b; Wu et al., 2025; Xiao et al., 2025; Liu et al., 2025a). Despite these advancements, achieving robust and powerful multimodal reasoning remains an open challenge.

## 3 METHOD

This section first details the Reasoning-Rendering-Visual-Feedback (RRVF) framework and then describes the reinforcement learning strategy used for its optimization.

## 3.1 RRVF FRAMEWORK

The core of RRVF is a closed-loop system comprising three key components: an iterative visual reasoner, a visual feedback mechanism, and a final visual judge.

### 3.1.1 VISUAL REASONING.

The process for generating rendering code is an iterative procedure, formally outlined in Algorithm 1. It begins with a single image input. In each turn, as illustrated in Figure 2(b), the MLLM produces a response (Line 3) that includes both internal reasoning in `<think>` tags and a specific action. For intermediate turns, this action is a code snippet enclosed in a `<tool_call>` tag. A call

is then made to an external tool to execute this code (Line 10). The resulting feedback is appended to the conversation history to inform the next turn. This iterative cycle concludes when the model's response contains the final solution within an `<answer>` tag, which terminates the process (Lines 5-7). The process also stops if a predefined maximum number of turns is reached (Madaan et al., 2023).

### 3.1.2 VISUAL FEEDBACK.

The visual feedback mechanism (Bai et al., 2022), illustrated in Figure 2(c), is a critical tool that provides structured guidance for the MLLM. Its operation involves two main steps:

- **Step 1: Rendering.** The code generated by the MLLM is executed by a specific rendering engine based on the domain. For data charts, Python code is executed using libraries such as Matplotlib. For web interfaces, the Playwright library is utilized to render the HTML in a browser and capture a screenshot. In cases where the code is invalid and rendering fails, the tool returns an explicit failure signal.

- **Step 2: Comparison and Feedback Generation.** The successfully rendered image is compared to the original input image. This comparison is performed by a separate, powerful MLLM acting as a qualitative assessor. It is prompted to articulate the specific visual discrepancies (e.g., colors, elements, missing text) in natural language. This descriptive feedback provides actionable information for the primary MLLM's next reasoning step.

### 3.1.3 VISUAL JUDGE.

The Visual Judge (Figure 2(d)) provides a quantitative score for the final output, which is essential for policy optimization. The process first renders the final code and then uses a powerful MLLM to evaluate the similarity between the generated and target images (Zheng et al., 2023). The MLLM-based score serves as one of the reward signals for reinforcement learning and forms the primary optimization objective.

## 3.2 REINFORCEMENT LEARNING OPTIMIZATION

For `image-to-code` generation tasks, the iterative and goal-oriented characteristics of the proposed RRVF framework render it particularly amenable to reinforcement learning formulations (Ouyang et al., 2022; Chen et al., 2021).

The optimization process can be effectively formulated as learning an optimal policy that generates accurate rendering code, thereby enhancing the model's fundamental visual understanding capabilities. Additionally, Group Relative Policy Optimization (GRPO) (Shao et al., 2024b) has been demonstrated to effectively guide model learning toward optimal trajectories through objective verification. Therefore, we employ GRPO as the optimization algorithm for RRVF to learn the capability of modeling rendering code by *verifying the visual similarity between rendered outputs and source images*. In the following sections, we provide a detailed description of our optimization algorithm.

### 3.2.1 OPTIMIZATION WITH GRPO.

The MLLM's policy, denoted as $\pi_\theta$, is optimized using GRPO, which serves as a computationally efficient alternative to Proximal Policy Optimization (PPO) (Schulman et al., 2017). Unlike PPO, GRPO eliminates the need for a separate value function for advantage estimation, thereby avoiding high computational overhead and streamlining the optimization process by generating a set of candidate outputs from the previous policy iteration.

Specifically, GRPO samples a group of outputs $\{o_1, o_2, \ldots, o_G\}$ for each input query $q$ from the old policy $\pi_{\theta_{\text{old}}}$. The method then optimizes the policy by maximizing a surrogate objective that incorporates clipping for stability, along with a KL regularization term to prevent excessive deviation from a reference policy $\pi_{\text{ref}}$. The objective function is given as:

$$J_{\text{GRPO}}(\theta) = \mathbb{E}_{q \sim P(Q),\, \{o_i\}_{i=1}^G \sim \pi_{\theta_{\text{old}}}(O|q)} \left[ \frac{1}{G} \sum_{i=1}^G \left( \min\left( \frac{\pi_\theta(o_i \mid q)}{\pi_{\theta_{\text{old}}}(o_i \mid q)} A_i, \right.\right.\right.$$
$$\left.\left.\left. \text{clip}\left( \frac{\pi_\theta(o_i \mid q)}{\pi_{\theta_{\text{old}}}(o_i \mid q)}, 1 - \varepsilon, 1 + \varepsilon \right) A_i \right) - \beta\, D_{\text{KL}}\big( \pi_\theta \,\|\, \pi_{\text{ref}} \big) \right) \right], \tag{1}$$

The advantage $A_i$ is derived as a normalized relative measure: $A_i = \frac{r_i - \text{mean}(\{r_1, r_2, ..., r_G\})}{\text{std}(\{r_1, r_2, ..., r_G\})}$, using the rewards from the sampled group to establish a baseline. Additionally, the KL divergence term $D_{\text{KL}}(\pi_\theta \| \pi_{\text{ref}})$ promotes controlled policy updates, balancing exploration and stability throughout training.

### 3.2.2 HYBRID REWARD DESIGN.

As depicted in Figure 2(e), a hybrid reward function is engineered to provide a comprehensive learning signal. Our reward design necessitates the provision of dense and accurate reward signals to ensure format compliance during the model's inference process while encouraging tool utilization to enhance the model's performance ceiling. To address these objectives, we design the following three reward components:

- **Visual Similarity Reward** ($R_{\textbf{vision}}$)**:** This is the primary reward component, sourced directly from the Visual Judge. It quantifies the fidelity between the final rendered image and the original input, providing a dense and accurate signal. This reward can be efficiently computed using an MLLM as a judge, thereby avoiding heavy reliance on curated image-text supervision.

- **Format Correctness Reward** ($R_{\textbf{format}}$)**:** This reward penalizes structural and syntactic errors. A penalty is applied if the reasoning, tool_call, and answer steps are improperly formatted or if the generated code is non-executable.

- **Tool-Use Reward** ($R_{\textbf{tool}}$)**:** This reward is designed to encourage tool use. It provides a small reward for each successful tool call, incentivizing the model to use the feedback loop to refine its answer. To avoid rewarding excessively long conversations, the reward is capped, balancing the need for exploration with the goal of convergence.

Furthermore, during RRVF training, we employ a weighted combination to balance these reward functions. The total reward $R$ for a completed trajectory is a weighted sum of three distinct components,

$$R = w_v R_{\text{vision}} + w_f R_{\text{format}} + w_t R_{\text{tool}}, \tag{2}$$

where $w_v, w_f$, and $w_t$ are hyperparameters that balance the contribution of each component.

## 4 EXPERIMENTS

### 4.1 DATASETS AND EVALUATION

Details of the datasets and evaluation metrics are provided in Appendix A.

A comprehensive set of both closed-source and open-source models is selected for comparison to ensure a thorough evaluation. The closed-source baselines include leading proprietary models: GPT-4o (OpenAI, 2024a), OpenAI o3 (OpenAI, 2025a), Gemini-2.5-Pro (Google, 2025), and Claude-4-Sonnet (Anthropic, 2025). The open-source counterparts include vision-language models such as LLaVA-OneVision (Li et al., 2024), Qwen2.5-VL (Bai et al., 2025), and InternVL3 (Zhu et al., 2025).

A comparison with supervised fine-tuning approaches is presented in the ablation study 4.5.

| Model | Exec rate | Text | Layout | Type | Color | GPT-4o score | Overall |
|---|---|---|---|---|---|---|---|
| *Closed-Source MLLMs* | | | | | | | |
| (2024/02) Gemini-1.0-Pro-Vision | 68.2* | 52.6* | 64.2* | 51.3* | 47.1* | 53.3* | 53.6* |
| (2024/11) GPT-4o-2024-11-20 | 90.00 | 66.55 | 79.31 | 71.83 | 60.84 | 82.50 | 76.06 |
| (2025/04) OpenAI o3 | 90.17 | 74.17 | 80.58 | 71.37 | 63.74 | 86.45 | 79.46 |
| (2025/05) Claude-4-Sonnet | 91.83 | 68.87 | 82.43 | 67.13 | 57.59 | 85.46 | 77.23 |
| (2025/06) Gemini-2.5-Pro | 93.33 | 84.95 | 83.37 | 75.05 | 66.90 | 90.58 | 84.07 |
| *Open-Source MLLMs* | | | | | | | |
| (2025/02) Qwen2.5-VL-72B-Instruct | 83.83 | 34.44 | 61.71 | 45.49 | 35.12 | 50.41 | 47.30 |
| (2024/03) DeepSeek-VL-7B | 41.3* | 15.3* | 26.6* | 19.7* | 14.5* | 20.4* | 19.7* |
| (2025/02) LLaVA-OneVision-7B | 17.28 | 7.97 | 13.55 | 9.15 | 7.36 | 10.01 | 9.76 |
| (2025/02) Qwen2.5-VL-7B-Instruct | 68.83 | 30.01 | 55.79 | 36.50 | 26.91 | 39.04 | 38.17 |
| (2025/04) InternVL3-8B | 71.67 | 45.03 | 57.89 | 45.87 | 38.88 | 54.91 | 50.91 |
| SFT [with text labels] | 69.00 | 56.97 | 63.60 | **60.53** | **51.89** | 62.09 | 60.17 |
| $\Delta$ (vs Qwen2.5-VL-7B-Instruct) | +0.17 | +26.96 | +7.81 | +24.03 | +24.98 | +23.05 | +22.00 |
| RRVF (Ours) [without text labels] | **97.83** | **62.47** | **80.97** | 53.56 | 46.41 | **67.87** | **64.36** |
| $\Delta$ (vs Qwen2.5-VL-7B-Instruct) | +29.00 | +32.46 | +25.18 | +17.06 | +19.50 | +28.83 | +26.19 |

Table 1: Performance comparison on the ChartMimic benchmark. We report the metrics from the original ChartMimic benchmark (Yang et al., 2024). The best and second-best results among open-source models under 10B parameters are **bolded** and underlined, respectively. Results marked with * are reported by the original benchmark.

| Model | Exec Rate | Text | GPT-4o Score | $Text_{pass}$ | $GPT\text{-}4o\ Score_{pass}$ |
|---|---|---|---|---|---|
| *Closed-Source MLLMs* | | | | | |
| (2023/09) GPT-4V | 84.1* | 48.53* | 5.45* | *57.7** | *6.48** |
| (2024/02) Gemini-1.0-Pro-Vision | 68.2* | 36.56* | 3.45* | *53.6** | *5.06** |
| (2024/06) Claude-3-Sonnet | 75.8* | 35.40* | 4.08* | *46.7** | *5.38** |
| (2024/11) GPT-4o-2024-11-20 | 90.15 | 48.91 | 6.09 | *54.25* | *6.76* |
| (2025/04) OpenAI o3 | 87.12 | 57.65 | 6.70 | *66.17* | *7.69* |
| (2025/05) Claude-4-Sonnet | 92.42 | 56.86 | 6.16 | *61.52* | *6.76* |
| (2025/06) Gemini-2.5-Pro | 87.88 | 71.70 | 7.65 | *81.59* | *8.71* |
| *Open-Source MLLMs* | | | | | |
| (2025/02) Qwen2.5-VL-72B-Instruct | 83.33 | 56.74 | 5.79 | *68.09* | *6.95* |
| (2024/03) Mini-Gemini-8x7B-HD | 73.5* | 29.91* | 2.84* | *40.7** | *3.87** |
| (2025/02) LLaVA-OneVision-7B | 84.09 | 26.72 | 2.75 | *31.78* | *3.27* |
| (2025/02) Qwen2.5-VL-7B-Instruct | 70.46 | 35.80 | 3.40 | *50.81* | *4.82* |
| (2025/04) InternVL3-8B | 76.52 | 30.67 | 3.25 | *40.08* | *4.25* |
| SFT [with text labels, ChartMimic trained] | 49.24 | 21.63 | 2.47 | *43.93* | *5.02* |
| $\Delta$ (vs Qwen2.5-VL-7B-Instruct) | -21.22 | -14.17 | -0.93 | - | - |
| RRVF (Ours) [without text labels] | **96.21** | **39.89** | **4.44** | *41.46* | *4.61* |
| $\Delta$ (vs Qwen2.5-VL-7B-Instruct) | +25.75 | +4.09 | +1.04 | - | - |

Table 2: Performance comparison on the python_matplotlib subset of the Plot2Code benchmark. The best and second-best results on the primary metrics (Exec Rate, Text, GPT-4o Score) among open-source models under 10B parameters are **bolded** and underlined, respectively. Results marked with * are reported by the original benchmark.

## 4.2 EXPERIMENTAL SETTINGS

### 4.2.1 TRAINING SETUPS.

The policy model is initialized from Qwen2.5-VL-Instruct-7B (Bai et al., 2025) and trained on a cluster of 8 NVIDIA A100 GPUs. The model is optimized using the GRPO algorithm with a global batch size of 32 and a learning rate of $1 \times 10^{-6}$. During each optimization step, a group of $G = 8$ candidate trajectories is sampled for each prompt. The RRVF interaction loop is set to a maximum of 4 turns, and the maximum sequence length for generation is 16,384 tokens. The KL divergence coefficient is set to 0.0.

Visual feedback and judgment are provided by a more capable model, Qwen2.5-VL-Instruct-72B (Bai et al., 2025), which serves as the reward oracle. To handle the substantial computational demand during the policy rollout phase, this judge model is deployed on a separate, dedicated cluster of 8 NVIDIA A100 GPUs. Inference is accelerated using the vLLM framework (Kwon et al., 2023).

The optimization is guided by the hybrid reward function from Equation 2. Following established practicesZheng et al. (2025), the weights are set to $w_v = 0.8$, $w_f = 0.2$, and $w_t = 1.0$. The interaction loop operates for a maximum of $T_{max} = 4$ turns. Within this loop, the tool-use efficiency reward, $R_{tool}$, assigns a reward of 1/3 for each successful tool call and is fixed to its maximum value of 1.0 if the visual similarity score surpasses 0.95.

Details of the cold-start and the utilized prompts are provided in Appendix B and Appendix F.

### 4.2.2 INFERENCE SETUPS.

During the evaluation phase, the model operates in a direct, single-turn inference mode. The final output is generated from the input prompt in a single pass, without access to the iterative refinement mechanism or any external tools available during the training loop. This protocol is intentionally designed to rigorously assess the model's internalized multimodal comprehension and generation capabilities, demonstrating the improvements gained autonomously after the completion of training.

Consequently, the inference latency remains identical to that of the standard base model, introducing zero computational overhead at test time. By strictly prohibiting tool invocation during inference, we ensure that the performance gains stem solely from the model's learned ability to internalize the rendering logic, rather than from external aids or extended test-time compute.

### 4.3 MAIN RESULTS

The main experimental results are presented in Table 1, Table 2, and Table 3. The findings indicate that the RRVF-trained model achieves SOTA performance among open-source MLLMs of comparable size. Case studies are provided in Appendix C.

### 4.3.1 CHART-TO-CODE TASK.

On the ChartMimic benchmark (Table 1), the RRVF-trained model exhibits substantial improvements over its base model, Qwen2.5-VL-7B-Instruct. The most notable result is the code execution rate (Exec rate) of 97.83%, which surpasses all other evaluated models, including proprietary systems like OpenAI o3 and Gemini-2.5-Pro. This exceptional reliability in generating valid code is a direct result of the RL process, which penalizes non-executable outputs. Consequently, the model attains an Overall score of 64.36, securing the highest performance among all open-source models. This performance is further supported by a strong score in layout understanding (Layout: 80.97), suggesting that the visual feedback loop effectively teaches the model to infer and replicate the underlying structural logic of data visualizations.

Table 3: Performance comparison on the Web-Sight benchmark. The best results among open-source models under 10B parameters are **bolded**.

| Model | CLIP | GPT |
|---|---|---|
| *Closed-Source MLLMs* | | |
| GPT-4o-2024-11-20 | 88.94 | 94.55 |
| OpenAI o3 | 91.58 | 96.49 |
| Claude-4-Sonnet | 92.30 | 96.46 |
| Gemini-2.5-Pro | 77.83 | 75.88 |
| *Open-Source MLLMs* | | |
| LLaVA-OneVision-7B | 79.74 | 72.61 |
| Qwen2.5-VL-7B-Inst | 83.50 | 84.17 |
| InternVL3-8B | 84.17 | 85.54 |
| **RRVF (Ours)** | **88.29** | **91.50** |

On the Plot2Code dataset (Table 2), the RRVF model achieves the highest execution rate (96.21%) among all models. Consequently, RRVF surpasses similarly sized models on the primary Text and GPT-4o Score metrics, underscoring its enhanced robustness and reliability. When considering "pass" scores, which are calculated only on successfully executed code, models with lower execution rates are effectively evaluated on easier subsets of data. Despite RRVF's high execution rate exposing it to more challenging examples, its performance remains highly competitive.

### 4.3.2 WEB-TO-CODE TASK.

The applicability of RRVF is further tested on the WebSight benchmark, a domain characterized by high structural freedom (Table 3). In this task, our RRVF-trained model achieves strong performance with a CLIP score of 88.29 and a GPT score of 91.50, surpassing similarly sized models. Crucially, this advantage is obtained without any access to ground-truth HTML, relying instead on our proposed method. This demonstrates that the framework can effectively learn to deconstruct and replicate complex layouts and component relationships from raw pixels alone, validating the potential of learning through pure visual feedback even in highly unstructured domains.

## 4.4 TRAINING ANALYSIS

A cold-start phase using LoRA-based Supervised Fine-Tuning is first conducted to familiarize the model with the tool-calling format, as detailed in Appendix B. This phase involves training for only one epoch, requiring approximately 15 minutes. In the subsequent reinforcement learning phase, the model performance is observed to stabilize after 100 steps, resulting in a total training duration of approximately 42 hours.

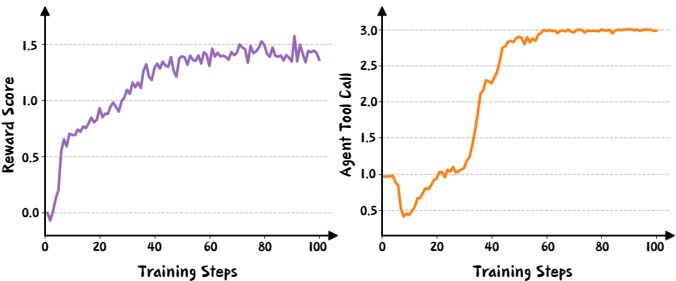

Figure 3 illustrates the training process, where a steadily increasing reward score confirms policy improvement. The number of tool calls follows a distinct dip-rise-plateau trend, aligning with trends reported in prior work (Zheng et al., 2025), reflecting the agent's progression from learning the tool's mechanics to fully exploiting iterative feedback.

Figure 3: Training curves for reward score and tool usage.

## 4.5 ABLATION STUDY

### 4.5.1 RRVF VS. SFT.

Our ablation study, conducted on the ChartMimic dataset, directly pits RRVF against a standard Supervised Fine-Tuning (SFT) baseline. Although SFT benefits from direct access to ground-truth code, RRVF demonstrates marked superiority. As shown in Table 1, RRVF achieves a near-perfect execution rate of 97.83%, significantly outperforming SFT's 69.00%. This highlights the reliability of the code generated by our framework. Furthermore, RRVF's dominant Layout score (80.97 vs. 63.60) confirms its advanced comprehension of holistic visual structures. This indicates that learning from the final visual outcome, as RRVF does, cultivates a more robust and compositional reasoning ability compared to the surface-level mimicry of SFT. A more profound finding is that our final 7B model, trained via RRVF, surpasses its much larger 72B teacher model in the Overall score (64.36 vs. 47.30). This counter-intuitive result proves that RRVF is more than knowledge distillation; it enables the policy model to actively explore the solution space and discover policies that are superior to its own feedback provider, thus bootstrapping its capabilities beyond initial constraints.

### 4.5.2 GENERALIZATION ABILITY.

**Dataset-Level Generalization** To evaluate the true extent of the learned skills, we test the models' generalization ability via a zero-shot evaluation on the unseen Plot2Code dataset. As shown in Table 2, the SFT-trained model's execution rate dramatically plummets from 69.00% on the ChartMimic to 49.24% on the Plot2Code. This sharp decline suggests that SFT learns brittle, non-transferable knowledge tied to the specific code patterns of the ChartMimic dataset. In stark contrast, our RRVF-trained model showcases exceptional generalization. Its execution rate remains remarkably stable, decreasing only negligibly from 97.83% to 96.21%. This stability is strong evidence that RRVF learns the fundamental and transferable principles of translating visual elements into programmatic logic, rather than simply memorizing code templates.

**Task-Level Generalization** We further evaluate zero-shot performance on diverse benchmarks (MathVerseZhang et al. (2024), MathVistaLu et al. (2023), LogicVistaXiao et al. (2024)) to assess general reasoning capabilities. As shown in Table 4, SFT suffers from consistent performance drops (e.g., 49.2 to 45.7 on MathVerse), a clear sign of catastrophic forgetting. Conversely, RRVF maintains performance on par with the base model across all tasks. This confirms that our method achieves domain specialization without compromising the model's foundational multimodal capabilities.

Table 4: Evaluation on general visual reasoning benchmarks.

| Model | MathVerse | MathVista | LogicVista |
|---|---|---|---|
| Qwen2.5-VL-7B | 49.2 | 68.2 | **44.1** |
| SFT [with text labels, ChartMimic trained] | 45.7 | 65.0 | 41.2 |
| RRVF[without text labels] | **49.5** | **68.9** | 43.9 |

### 4.5.3 Impact of Iterative Reasoning Turns

The necessity of the multi-turn self-correction mechanism is assessed by varying the maximum reasoning turns ($T_{max}$) during training. All results are reported on the ChartMimic benchmark, and all other training and inference settings are kept identical to the main experiments. As shown in Table 5, when a single turn is enforced ($T_{max} = 1$), the visual feedback loop is disabled and a notable degradation

Table 5: Ablation on Iteration Turns ($T_{max}$).

| $T_{max}$ | Exec Rate | Overall |
|---|---|---|
| 1 | 85.53% | 56.21 |
| 4 | **97.83%** | **64.36** |

is observed, with the execution rate decreasing to 85.53%. When up to four turns are allowed ($T_{max} = 4$), the code is iteratively refined against visual discrepancies, and the execution rate is increased to 97.83% while the Overall Score reaches 64.36. These results indicate that the iterative "Reasoning, Rendering and Visual Feedback" loop is crucial for complex visual reasoning that is not solvable in a single pass.

Further ablation studies investigating the framework's robustness across different base models and the impact of individual reward components are provided in Appendix D and Appendix E.

## 5 Conclusions

This paper presents RRVF, a framework that enhances visual reasoning in multimodal large language models through a reasoning-rendering-visual-feedback loop. The framework leverages the "Asymmetry of Verification" principle to learn generative logic directly from raw images, eliminating the need for paired textual supervision. Comprehensive experiments across charts and web interfaces show that RRVF consistently outperforms strong open-source baselines. The ablation study further confirms the superiority of the proposed framework over traditional supervised fine-tuning, validating its effectiveness in fostering robust visual reasoning.

ETHICS STATEMENT

Our work utilizes publicly available datasets (ChartMimic, Plot2Code, WebSight) and open-source models (Qwen2.5-VL). The research does not involve human subjects, private data, or create applications with immediate potential for harm.

REPRODUCIBILITY STATEMENT

We are committed to ensuring the reproducibility of our work. All code for the RRVF framework, training scripts, and evaluation protocols will be made publicly available in a GitHub repository upon publication. The repository will include detailed instructions for setting up the environment, running the training, and reproducing the results reported in this paper.

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

## A DATASETS AND METRICS.

The framework's performance is evaluated on two `image-to-code` tasks: `chart-to-code` and `web-to-code`. For the `chart-to-code` task, two standard benchmarks are utilized. The official "chart-to-code" subset of the ChartMimic dataset (Yang et al., 2024) comprises 2,400 test images. From this set, 1,800 images are repurposed for training, while the remaining 600 images, which form the official smaller test set, are used for evaluation. Additionally, zero-shot evaluation is performed on the `python_matplotlib` subset of the Plot2Code dataset (Wu et al., 2024a), which comprises 132 test cases. For the `web-to-code` task, only 2,000 screenshots are randomly sampled from the WebSight dataset (Laurençon et al., 2024) for training, with a disjoint set of 500 images reserved for testing.

For the `chart-to-code` task, evaluation adheres to the official protocols and metrics of their respective benchmarks. For the `web-to-code` task, however, traditional metrics such as block-level HTML comparisons are ill-suited for our framework. This is because the model is designed to generate semantically equivalent but structurally distinct code, without access to the ground-truth source. Consequently, evaluation for this task is based on visual fidelity and semantic correctness, quantified using the following metrics. (1) CLIP Similarity: The perceptual similarity between the rendered image and the original input is measured by the cosine similarity of their respective CLIP embeddings (Radford et al., 2021). (2) LLM-based Assessment: A powerful multimodal model, GPT-4o (OpenAI, 2024a), is employed as an impartial judge to assess the quality of the generated outputs. It evaluates semantic consistency, element completeness, and overall visual fidelity against the input image.

A comparison with conventional supervised fine-tuning approaches is presented in the ablation study.

## B COLD START DETAILS

In experimental studies, tasks across different modalities exhibit varying sensitivities to the model's learning paradigm. Specifically, the Qwen2.5VL-7B-Instruct model, when applied to the web_to_code tasks, converges directly to the expected performance using RRVF training framework, without requiring a dedicated cold-start phase.

However, the chart_to_code task presents a significant challenge. We observed that without a cold-start, the model generates a high frequency of format errors when attempting to make tool calls. This phenomenon triggers a vicious cycle: the more the model attempts to call the tool, the more likely it is to produce malformed outputs, leading to persistent failures and severely impeding the model's ability to learn effective strategies.

To address tool-calling failures in the chart_to_code task, a cold-start dataset is synthesized using a score-guided approach. The process begins with 2,500 images from the ChartQA training set, which is chosen for its simpler chart types to isolate the learning of the tool-calling syntax from the complexity of the chart itself. Multi-turn dialogues are generated for these images using GPT-4o, and any instances with format errors are discarded. The remaining valid data is then partitioned into a four-stage curriculum based on turn-by-turn reward scores:

- **Stage 1** (1-Turn, 191 samples): Samples ranking in the top 10% by first-turn score.

- **Stage 2** (2-Turn, 573 samples): From the remainder, samples ranking in the top 30% by second-turn score.

- **Stage 3** (3-Turn, 765 samples): From the remainder, samples ranking in the top 40% by third-turn score.

- **Stage 4** (4-Turn, 384 samples): The remaining complex cases.

This stratification process results in a final cold-start dataset of 1,913 dialogues. The average dialogue length is approximately 5,700 tokens, providing substantial data for learning the complex tool-call protocol.

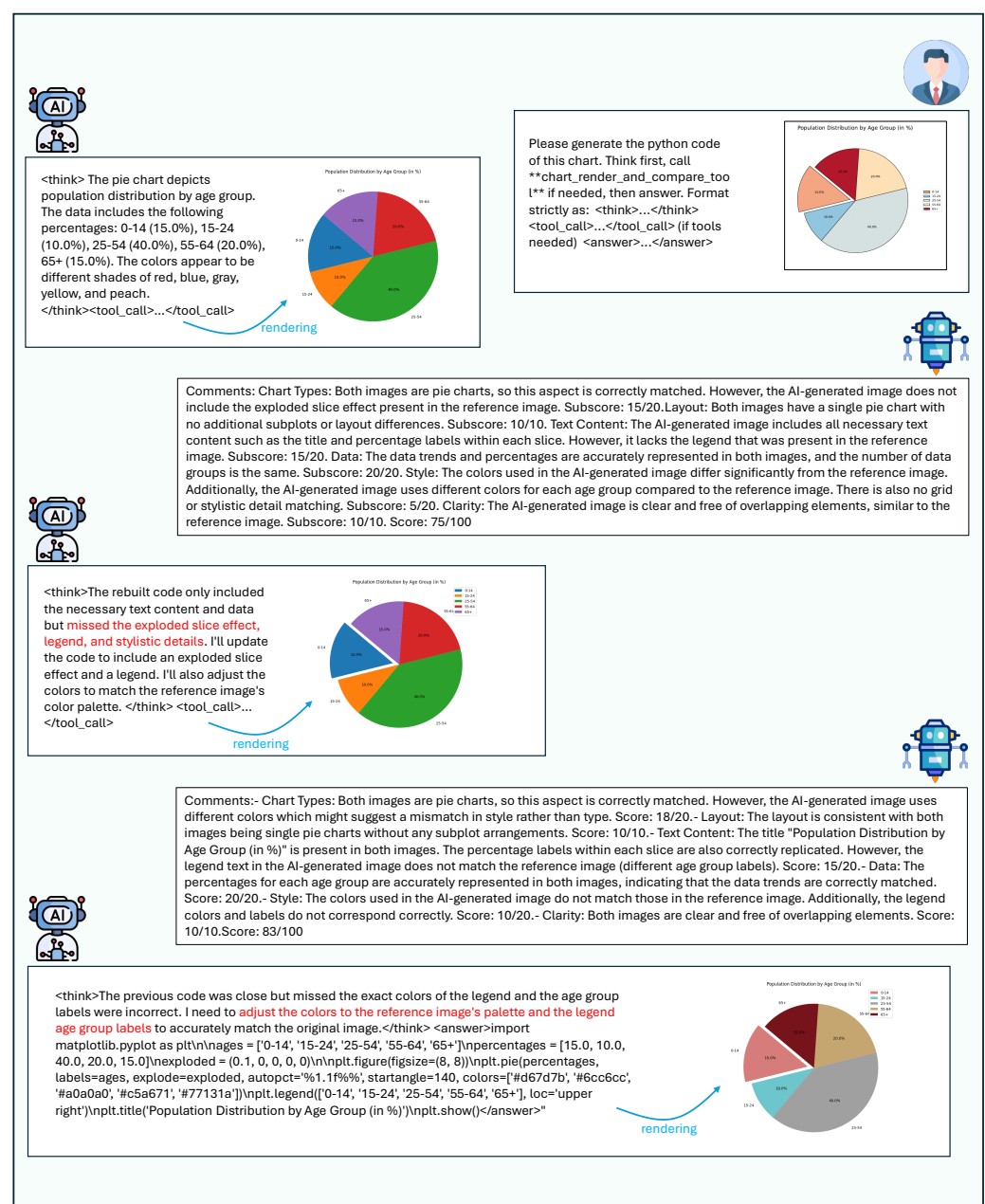

Figure 4: An illustration of the RRVF training process. The code for a pie chart is iteratively refined based on visual feedback. **(Turn 1)** A basic pie chart is generated but lacks the required exploded slice effect. **(Turn 2)** Following feedback, the code is updated to incorporate the effect and a legend, though color discrepancies persist. **(Turn 3)** The color palette and legend are adjusted to align with the reference image. The progressive refinement is reflected in the increasing score ($75 \rightarrow 83 \rightarrow 85$), demonstrating the efficacy of the closed-loop learning process.

## C  CASE STUDY

Figure 4 presents a case study of the learning mechanism, where the generated output for a pie chart is progressively refined. In the initial turn, the code produces a basic pie chart, but feedback indicates it "does not include the exploded slice effect present in the reference image." In response, the code is updated in the second turn to add the effect and a legend. While this improves the structure, a

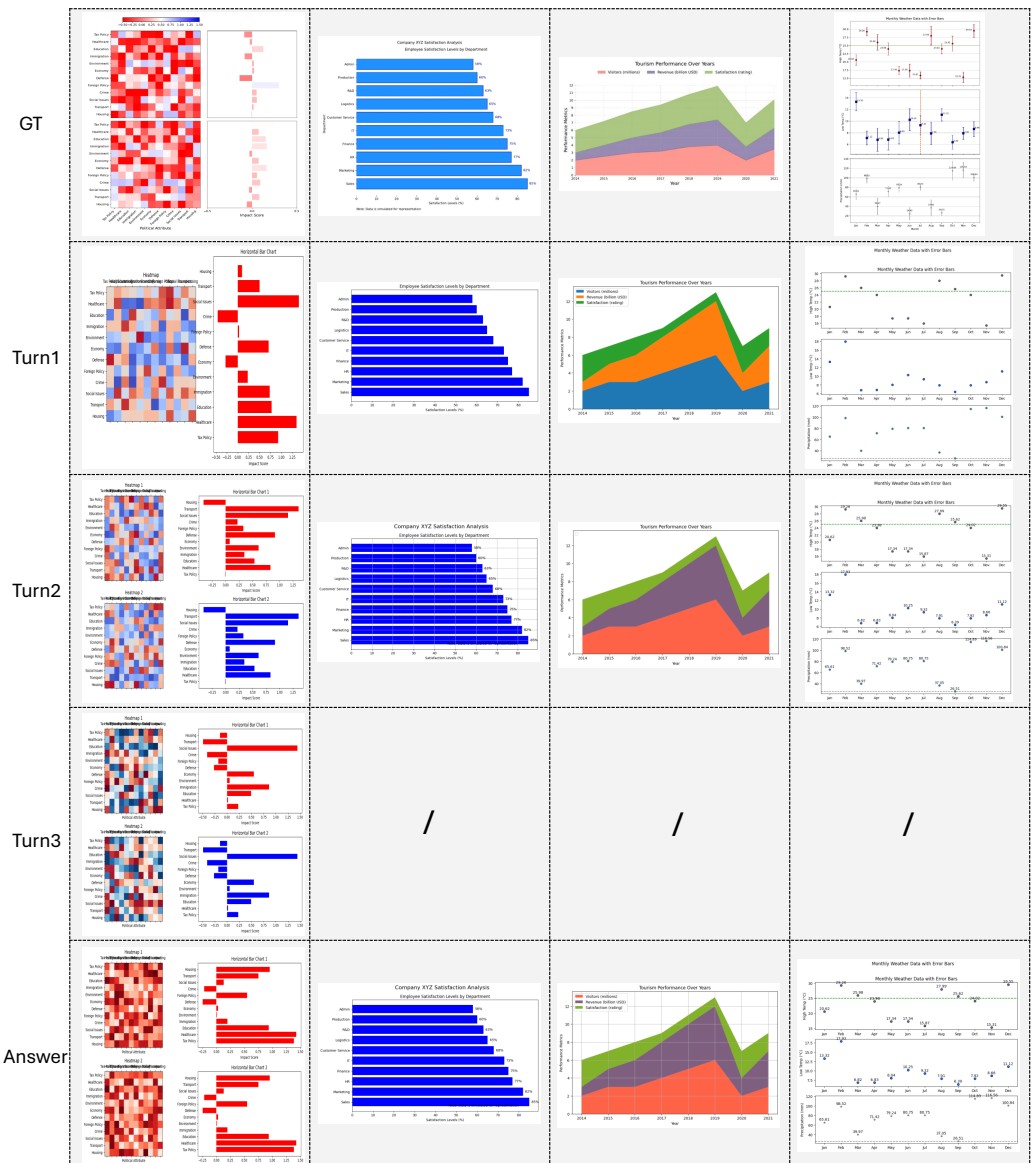

Figure 5: Training examples illustrating the generalizability of the RRVF framework. The framework demonstrates its capability to iteratively refine diverse chart types, including bar, line, and scatter plots. This is achieved by progressively correcting errors in structure, data, and style based on multi-turn visual feedback.

color mismatch is subsequently identified. The final turn corrects the color palette and legend labels, leading to an output that closely replicates the target image. This iterative correction is reflected in the increasing score ($75 \rightarrow 83 \rightarrow 85$).

Figure 5 presents several randomly sampled examples from training trajectories that achieved a final similarity score above 80. These cases demonstrate that the self-correction mechanism is effective across diverse chart types, including bar, line, and scatter plots. Analysis of the training rollouts reveals that this iterative process builds an internalized reasoning capability. This capability is fundamental to the model's enhanced performance during evaluation, where it operates in a single pass without tool acces

## D    EFFECTIVENESS ACROSS DIFFERENT BASE MODELS

The efficacy of RRVF across base models is examined using Qwen2-VL-7B and Qwen2.5-VL-7B as base models on ChartMimic, with all other settings aligned with the main experiments. Consistent gains with RRVF are observed across both backbones, as summarized in Table 6. With Qwen2-VL-7B, the execution rate is 92.32% and the Overall Score is 61.65. With Qwen2.5-VL-7B, higher absolute performance is obtained (execution rate 97.83%, Overall Score 64.36), while the relative improvements are observed to be consistent. These findings support the robustness and broad applicability of the proposed RRVF.

Table 6: Generalizability Across Base Models (with RRVF).

| Base Model | Exec Rate (%) | Overall Score |
|---|---|---|
| Qwen2-VL-7B + RRVF | 92.32 | 61.65 |
| Qwen2.5-VL-7B + RRVF | **97.83** | **64.36** |

## E    IMPACT OF REWARD COMPONENTS

To verify the design of our hybrid reward function, we conduct an ablation analysis by removing each component individually. (1) **w/o Format Reward**: The model struggles to adhere to strict syntax rules (e.g., XML tags), leading to a drastic drop in the execution rate. (2) **w/o Tool-Use Reward**: The policy becomes conservative, favoring short, single-turn responses and underutilizing self-correction mechanisms. (3) **w/o Visual Reward**: Code is executable but lacks semantic alignment, producing charts with incorrect content or blank layouts. These results confirm that all three components are essential: Format ensures executability, Visual guarantees correctness, and Tool-Use incentivizes exploration.

## F    PROMPT

This section details the prompt templates used for training and evaluation. These templates include system and user prompts for both the 'chart_to_code' task (Figures 6 and 7) and the 'web_to_code' task (Figures 8 and 9). Additionally, a prompt for the automated evaluation of 'web_to_code' outputs using GPT-4o is presented in Figure 10.

## G    USE OF LARGE LANGUAGE MODELS

Large Language Models are used for grammar check and polishing in this paper.

System prompt template for the chart_to_code task

```
You are a helpful assistant.

# Tools
You may call one or more functions to assist with the user query.
You are provided with function signatures within <tools></tools>
    ↪ XML tags:
<tools>
{"type":"function","function":{"name":"chart_render_and_compare_tool",
    ↪ "description":"Renders a chart from python code and compares
    ↪ it with the original chart image. It returns a description
    ↪ of the
    ↪ differences.","parameters":{"type":"object","properties":
    ↪ {"chart_code":{"type":"string","description":"The python
    ↪ code to generate the chart."}},"required":["chart_code"]}}}
</tools>

# How to call a tool
Return a json object with function name and arguments within
    ↪ <tool_call></tool_call> XML tags:
<tool_call>
{"name": <function-name>, "arguments": <args-json-object>}
</tool_call>

**Example**:
<tool_call>
{"name": "chart_render_and_compare_tool", "arguments":
    ↪ {"chart_code": """ + "import matplotlib.pyplot as
    ↪ plt\\nimport numpy as np\\nfig, ax =
    ↪ plt.subplots()\\nax.plot([1, 2, 3, 4], [1, 4, 2,
    ↪ 3])\\nplt.show()" + """}}
</tool_call>
```

Figure 6: System prompt template for the 'chart_to_code' task

User prompt template for the chart_to_code task

```
Please generate the python code of this chart. Think first, call
    ↪ **chart_render_and_compare_tool** if needed, then answer.
    ↪ Format strictly as: <think>...</think>
    ↪ <tool_call>...</tool_call> (if tools needed)
    ↪ <answer>...</answer>
```

Figure 7: User prompt template for the 'chart_to_code' task

System prompt template for the web_to_code task

```
You are a helpful assistant.

# Tools
You may call one or more functions to assist with the user query.
    ↪ You are provided with function signatures within
    ↪ <tools></tools> XML tags. This is a very long line to
    ↪ demonstrate the automatic line wrapping feature provided by
    ↪ the listings package.

<tools>
{"type":"function","function":{"name":
    ↪ "html_render_and_compare_tool", "description":"Renders a
    ↪ html from html code and compares it with the original html
    ↪ image. It returns a description of the differences.",
    ↪ "parameters":{"type":"object","properties":
    ↪ {"html_code":{"type":"string","description":"The html code
    ↪ to generate the html."}},"required":["html_code"]}}}
</tools>

# How to call a tool
Return a json object with function name and arguments within
    ↪ <tool_call> XML tags:
<tool_call>
{"name": <function-name>, "arguments": <args-json-object>}
</tool_call>

**Example**:
<tool_call>
{"name": "html_render_and_compare_tool", "arguments": {"html_code":
    ↪ """ + HTML_CODE_EXAMPLE + """}}
</tool_call>
```

Figure 8: System prompt template for the 'web_to_code' task.

User prompt template for the web_to_code task

```
Please generate the HTML code of this webpage, only HTML code,
    ↪ including necessary CSS. The HTML code should be as concise
    ↪ as possible. The HTML code should be as concise as possible.
    ↪ Think first, call **html_render_and_compare_tool** if
    ↪ needed, then answer. Format strictly as:  <think>...</think>
    ↪  <tool_call>...</tool_call> (if tools needed)
    ↪ <answer>...</answer>
```

Figure 9: User prompt template for the 'web_to_code' task.

Evaluation prompt template for the 'web_to_code' task using GPT-4o as an automated judge

```
Score the similarity between the AI-generated image(Image 2) and
    ↪ the reference image(Image 1).
NOTE: Per instructions, missing images and icons in the
    ↪ AI-generated image (Image 2) should be ignored and will not
    ↪ affect the score.

Evaluation Format:
---
Comments:
-Layout (30 points): ${comment and subscore}
-Text (40 points): ${comment and subscore}
-Style (30 points): ${comment and subscore}

Score: ${final score}/100
```

Figure 10: Evaluation prompt template for the 'web_to_code' task using GPT-4o as an automated judge.

