# OpenReview forum: "RRVF: Visual Reinforcement Learning with Reasoning, Rendering, and Visual Feedback"
_ICLR.cc/2026/Conference — Submitted to ICLR 2026_

### Official Review · Reviewer_Ztqe · 2025-10-30

**Soundness:** 3
**Presentation:** 3
**Contribution:** 3
**Rating:** 8
**Confidence:** 3

**Summary:**

This paper studies the image-to-code problem via "Reasoning-Rendering-Visual-Feedback". This framework does not require manually labeled text instructions. It simply renders the generated code and uses an MLLM to verify the original image and the rendered image. Experiments under various protocols demonstrate that the proposed method brings significant improvements over baselines.

**Strengths:**

1. This paper is overall well-written and easy to follow.
2. Both the motivation and the solution are quite clear and reasonable.
3. Improvements are quite significant.

**Weaknesses:**

I only have one minor concern:

1. The generalization of this method. The trained model is excellent at code generation. How about other reasoning-related benchmarks, e.g, mathvista, mathverse, logicvisa, etc. Do these advanced code generation capabilities implicitly contribute to better general reasoning capabilities?

**Questions:**

N/A.

---

> ### Author Response · Authors · 2025-11-20
> **Response**
>
> We sincerely appreciate your positive assessment and recognition of the contribution. We are encouraged that you find the motivation clear and the improvements significant. The response to your inquiry regarding generalization is provided below.
>
> ### [Response to Weakness 1] Generalization to broader reasoning benchmarks
>
> To investigate whether the advanced code generation capabilities contribute to or hinder general reasoning, additional zero-shot evaluations were conducted on three diverse benchmarks: **MathVerse, MathVista, and LogicVista**.
>
> | Model                | MathVerse | MathVista | LogicVista |
> |:---------------------|:---------:|:---------:|:----------:|
> | Qwen2.5-VL-7B (base) |   49.2    |   68.2    |    44.1    |
> | SFT (ChartMimic)     |   45.7    |   65.0    |    41.2    |
> | RRVF (ours)          |   49.5    |   68.9    |    43.9    |
>
> The results reveal an important finding:
> 1.  **SFT suffers from catastrophic forgetting:** Supervised Fine-Tuning on the specific code generation dataset leads to a noticeable performance drop on general reasoning tasks.
> 2.  **RRVF preserves general capabilities:** In contrast, the proposed framework maintains performance on par with (or slightly better than) the base model.
>
> This suggests that RRVF transforms the model into a domain specialist without compromising its general visual reasoning abilities. The logical rigor required for code generation appears to align well with the reasoning chains needed for general mathematical and logical tasks, preventing the degradation typically associated with fine-tuning.
>
> We are actively exploring larger-scale self-learning pipelines (scaling up the data synthesis and RL training loop) to determine if this "verification-driven" paradigm can yield emergent capabilities in even broader, unstructured scenarios. We believe this work serves as a solid stepping stone in that direction.

---

### Official Review · Reviewer_WLZB · 2025-11-01

**Soundness:** 2
**Presentation:** 2
**Contribution:** 2
**Rating:** 4
**Confidence:** 4

**Summary:**

This paper proposes a reinforcement learning framework named "Reasoning-Rendering-Visual-Feedback" that only utilizes image data rather than image-text supervision. The framework only utilizes outcome reward by the GRPO algorithm. The training framework enables models to have multi-turn reasoning and tool-call abilities. The experiments show that the framework brings more stable improvement than SFT on Qwen2.5-VL-7B.

**Strengths:**

1. The paper designs the understanding of icon-type images as a rendering and verifying process, which is an interesting and reasonable idea.

2. The paper designs the training of multimodal reasoning as a multi-round rendering and verifying process, so that the model only needs to rely on images and does not require additional text annotations. This alleviates the dependence of multimodal reasoning training on text annotations.

3. The authors' experiments show that the proposed framework brings more significant advantages than supervised training (SFT).

**Weaknesses:**

1. The proposed framework requires multiple rounds of inference and tool calls during both training and inference. How long does training take? How much does inference time increase compared to not using tools?

2. The applicability of this framework in visual inference is limited. It can only be used in chart-to-code and web-to-code scenarios. The authors did not explore how this framework adapts to general image inference.

3. In Tables 1 and 2, did the authors only conduct in-the-domain experiments on the ChartMimic and Plot2Code benchmarks? I think generalization experiments should be added to demonstrate that reinforcement learning algorithms like GRPO, in addition to performance improvements, also enhance the model's generalization ability for SFT.

4. The authors only conducted SFT and GRPO experiments on Qwen2.5-VL-7B-Instruct, without experimenting with other baseline models, failing to demonstrate the framework's generality.

5. The authors did not conduct ablation experiments on the proposed framework. For example, using different maximum rounds, removing certain proposed components, etc., can validate the effectiveness of the entire framework.

**Questions:**

1. Refer to the issues raised in the weakness section.
2. How many steps were trained for the reinforcement learning fine-tuning? The curve shown in Figure 3 only shows 100 training steps, which seems to indicate the possibility of overfitting.

---

> ### Author Response · Authors · 2025-11-20
> **Response [1/2]**
>
> We sincerely thank you for the constructive feedback and insightful comments. We have carefully addressed your concerns below.
>
>
> ### [Response to W1] Training Time & Inference Latency
>
> > Training Time:
>
> As detailed in the revised **Section 4.4**, total training requires approximately 42 hours on 8×NVIDIA A100 GPUs, encompassing both the cold-start initialization and the RL optimization phases.
>
> > Inference Latency:
>
> As detailed in the **Common Response**, we emphasize again that **no tools are invoked during inference** (Section 4.2.2). The model operates in single-pass mode, maintaining inference speed identical to the baseline without tools.
>
>
> ### [Response to W2 & W3] Applicability
>
> The RRVF framework is specifically assessed in domains characterized by a clear **rendering-verification loop**, such as chart-to-code and web-to-code. These domains were selected as they provide an ideal testbed for the paper's core hypothesis: examining whether MLLMs can learn complex visual-to-code mappings solely from visual feedback, leveraging the "Asymmetry of Verification" principle.
>
> Regarding the concerns about generalization, we address two key aspects: (i) dataset-level generalization (W3), and (ii) the impact on general visual reasoning capabilities (W2b).
>
> 1. Dataset-level generalization (W3):  In Tables 1 and 2, ChartMimic is used **only for training**, and Plot2Code is used **purely as a zero-shot, out-of-distribution test set**. The distribution gap between them is reflected by SFT: its execution rate drops from 69.00% (ChartMimic) to 49.24% (Plot2Code), showing strong overfitting to the training benchmark. In contrast, RRVF remains stable (97.83% → 96.21%), indicating that GRPO-based training captures transferable rendering logic rather than memorizing dataset-specific patterns.
>
> 2. General ability and task-level applicability (W2): To examine whether the framework harms or preserves general visual reasoning, additional zero-shot evaluations are conducted on broader multimodal reasoning benchmarks:
>
> | Model                | MathVerse | MathVista | LogicVista |
> |:---------------------|:---------:|:---------:|:----------:|
> | Qwen2.5-VL-7B (base) |   49.2    |   68.2    |    44.1    |
> | SFT (ChartMimic)     |   45.7    |   65.0    |    41.2    |
> | RRVF (ours)          |   49.5    |   68.9    |    43.9    |
>
> The results indicate that SFT suffers from catastrophic forgetting, with performance dropping. In contrast, **RRVF maintains performance**, demonstrating that it effectively learns domain-specific logic without compromising general visual reasoning capabilities.

---

> ### Author Response · Authors · 2025-11-20
> **Response [2/2]**
>
> ### [Response to W4] Choice of baseline models.
>
>
> To provide a more comprehensive evaluation and address the concern regarding baseline coverage, the performance of **Qwen2-VL-7B** was evaluated on the ChartMimic test set.
>
> The comparative results are presented below:
>
> | Base Model | Execution Rate | Overall Score |
> |:-----:|:---------:|:------:|
> |   Qwen2-VL-7B    |   92.32   |  61.65 |
> |   Qwen2.5-VL-7B    |   97.83   |  64.36 |
>
> These results show that consistent performance gains are achieved across different base models, and the method is generally effective but exhibits different sensitivity to different base models.
>
> ### [Response to W5] Ablation studies on framework components.
>
> We thank you for suggesting these ablation studies, as they significantly strengthen the validation of our framework. Additional ablations have been conducted and will be included in the revised version.
>
> To analyze the contribution of the iterative self-correction mechanism, the framework was trained with varying maximum turn limits T_max in {1,  4}. As shown in the table below, performance on the ChartMimic task improves as the allowed reasoning depth increases. The model trained with T_max=4 achieves a 97.83% execution rate, significantly outperforming the T_max=1 (85.53%). This confirms that the multi-turn feedback loop is essential for refining complex rendering logic.
>
> | T_max | Exec Rate | Overall |
> |:-----:|:---------:|:------:|
> |   1   |   85.53   |  56.21 |
> |   4   |   97.83   |  64.36 |
>
>
> The necessity of each component in the hybrid reward function was also analyzed:
>
> **No Format:** The model struggles to adhere to strict syntax rules (e.g., XML tags), leading to a drastic drop in the Execution Rate
>
> **No Tool-Use:** The policy becomes conservative, favoring short, single-turn responses and underutilizing self-correction mechanisms.
>
> **No Visual:** Code is executable but lacks semantic alignment, producing charts with incorrect content or blank layouts.
>
>
>
> ### [Response to Q2] Reinforcement learning steps and overfitting concerns.
>
>
> As detailed in revised Section 4.4, the model was trained for **100** steps. This training scale is consistent with recent works on visual reasoning RL [1, 2, 3]. Unlike general-purpose pre-training, our framework targets a specific domain (Image-to-Code). The solution space is more constrained, allowing for faster policy optimization. Moreover, the RRVF framework employs multi-turn interactions (reasoning → rendering → visual feedback). Each episode contains multiple rounds of self-correction, providing significantly denser supervision signals than standard single-turn RL. This rich information gain per step accelerates learning essentially, a benefit not captured by merely counting optimization steps.
>
> Regarding the overfitting concern, the evidence in the original experiments already demonstrates **dataset-level generalization** (e.g., from ChartMimic to Plot2Code), and the newly added experiments further support **task-level generalization**. Detailed discussions are provided in [Response to W2 & W3].
>
>
> ---
>
> [1] Zheng, Ziwei, et al. "DeepEyes: Incentivizing" Thinking with Images" via Reinforcement Learning." arXiv preprint arXiv:2505.14362 (2025).
>
> [2] Wang, Haozhe, et al. "Vl-rethinker: Incentivizing self-reflection of vision-language models with reinforcement learning." arXiv preprint arXiv:2504.08837 (2025).
>
> [3] Huang, Wenxuan, et al. "Vision-r1: Incentivizing reasoning capability in multimodal large language models." arXiv preprint arXiv:2503.06749 (2025).
>
>
>
>
> ---
> Once again, we thank you for your constructive comments, which have helped us significantly improve the paper. All newly added content has been incorporated into the updated revision and highlighted in RED.

---

> ### Author Response · Authors · 2025-11-26
> **Short Summary of Discussion**
>
> Dear Reviewer WLZB, Thank you for your insightful suggestions. As the discussion period is coming to a close and our detailed response has been posted for a week, we kindly ask you to review our updates addressing your concerns :) .
>
> **(1) Training Time & Inference Latency:** We clarified that inference is **tool-free** (single-pass mode) with no overhead. We also reported the full training budget(≈42 hours on 8×A100).
>
> **(2) Applicability & Dataset-Level Generalization:** We validated zero-shot transfer from ChartMimic to Plot2Code, where RRVF remains stable while SFT drops significantly. We also added evaluations on **MathVerse/MathVista/LogicVista**, confirming RRVF preserves general reasoning without catastrophic forgetting.
>
> **(3) Basemodel Coverage:** We added experiments with **Qwen2-VL-7B**, demonstrating consistent improvements across different base models.
>
> **(4) Ablation Studies:** We implemented ablations on **reasoning turns** (T_max ∈ {1,4}) and **reward components** (Format/Tool-Use/Visual). The results quantitatively validate the necessity of multi-turn self-correction and the hybrid reward design.
>
> **(5) Training Steps & Overfitting:** We justified the 100-step training scale by citing recent visual reasoning RL works and highlighting the dense information from multi-turn interactions. Generalization results on Plot2Code and general benchmarks confirm no overfitting.
>
>
> **We would be very grateful if you could let us know whether they adequately resolve your concerns, or if there is anything further we can clarify to support your final decision.**

---

### Official Review · Reviewer_9ChV · 2025-11-06

**Soundness:** 2
**Presentation:** 3
**Contribution:** 2
**Rating:** 4
**Confidence:** 4

**Summary:**

The paper proposes a novel paradigm for multimodal reasoning training by eliminating the need for paired image-text/code supervision. Instead, the model learns solely from raw images via iterative rendering and visual comparison. This formulation shifts the paradigm from supervised imitation learning to self-supervised verification-driven RL. The authors introduce a closed-loop reasoning framework that iteratively generates code, executes it, and incorporates visual feedback for refinement. This mechanism allows the model to progressively correct its outputs based on rendered results rather than relying solely on single-pass generation.

**Strengths:**

1. The paper is well-written and easy to follow.

2. The research direction is impactful because collecting high-quality program annotations for visual tasks is expensive and often subjective.

3. The model achieves a high code execution rate on ChartMimic and performs competitively on WebSight, without requiring paired text supervision. It also shows better performance than supervised fine-tuning and comparable open-source baselines.

**Weaknesses:**

1. The framework depends on multi-round generation and external tool execution during training and inference. This raises questions regarding computational cost, latency, and scalability. The paper does not report training time, tool-call frequency, or inference speed compared to standard single-pass models.

2. The method is only evaluated on chart-to-code and web-to-code settings, which are structured scenarios with clearly defined rendering engines. It remains unclear whether the approach can extend to broader visual inference tasks (e.g., general scene understanding, reasoning, VQA).

3. The method exhibits a strong reliance on reward shaping and prompt design. The format reward, tool-use reward, and their weighting require manual tuning, yet the paper does not provide systematic analysis of reward stability or robustness. Additionally, no ablation or sensitivity study is conducted on the reward components, making it difficult to assess how much each part contributes to the final performance.

**Questions:**

1. Training cost & efficiency: How long does training take, and how many RL steps are used? What is the total computational budget?

2. Inference overhead: How much slower is inference when using the iterative tool-based framework? Is there a version that can operate efficiently without tool calls at test time?

3. Generalization evaluation: Did you conduct any experiments on tasks outside chart/web code generation? Could you add out-of-domain tests to verify generalization benefits of GRPO over SFT?

---

> ### Author Response · Authors · 2025-11-20
> **Response [1/2]**
>
> Thank you for the constructive feedback. Responses to specific concerns are provided below.
>
>
> ### [Concern 1 (Weakness 1 & Question 1 & Question2)] Training cost, efficiency, and inference overhead.
>
> > 1. Inference Overhead
>
> As clarified in the Common Response, the model uses **single-pass generation** without tools during inference. Therefore, the inference speed is close to that of the base model.
>
> > 2. Training Cost & Budget
>
> To address the request for specific computational details, **a detailed breakdown has been added to the revised Section 4.4**. The total training budget is approximately **42 hours on a cluster of 8 NVIDIA A100 GPUs**. This includes both the cold-start phase and the RL phase.
>
> > 3. RL Steps
>
> As illustrated in the **original Figure 3**, the model performance converges and stabilizes after approximately 100 optimization steps.
>
> > 4. Tool-Call Frequency
>
> During training, tool execution overhead is minimized through parallel concurrency and vLLM acceleration, **as originally described in Section 4.2.1**. Empirical monitoring confirms that tool invocation consumes less than 25% of the total time per step, ensuring high training throughput.
>
>
>
> ### [Concern 2 (Weakness 2 & Question 3)] Generalization to broader visual inference tasks
>
> To assess generalizability, we evaluated the model on three diverse benchmarks: MathVerse, MathVista, and LogicVista.
>
> | Model                | MathVerse | MathVista | LogicVista |
> |:---------------------|:---------:|:---------:|:----------:|
> | Qwen2.5-VL-7B (base) |   49.2    |   68.2    |    44.1    |
> | SFT (ChartMimic)     |   45.7    |   65.0    |    41.2    |
> | RRVF (ours)          |   49.5    |   68.9    |    43.9    |
>
> The results indicate that **SFT suffers from catastrophic forgetting**, with performance dropping. In contrast, **RRVF maintains performance on par with the base model**, demonstrating that it effectively learns domain-specific logic without compromising general visual reasoning capabilities.

---

> ### Author Response · Authors · 2025-11-20
> **Response [2/2]**
>
> ### [Concern 3 (Weakness 3)] Reward shaping and sensitivity
>
> We sincerely appreciate your careful attention to detail. We acknowledge a typographical error in the original manuscript regarding the reward weights in Section 4.2.1. The correct values are w_v = 0.8, w_f = 0.2, and w_t = 1.0. We apologize for any confusion this may have caused and have corrected these values in red text in the updated revision.
>
> The hybrid reward formulation is detailed in **Section 3.2.2 (Hybrid Reward Design)**. Rather than relying on heuristic manual tuning, the design is grounded in a principled logical hierarchy inspired by tool-based RL frameworks (e.g., DeepEyes [1]):
>
> 1.  Format Reward: Ensures basic executability. This serves as a prerequisite, as syntax errors render subsequent visual evaluation impossible.
>
> 2.  Visual Reward: Quantifies semantic correctness. This measures the alignment between the rendered output and the source image, functioning as the primary optimization objective.
>
> 3.  Tool-Use Reward : Encourages exploration. This prevents premature convergence to single-turn solutions and incentivizes iterative refinement.
>
> Empirical results suggest that performance remains stable against minor variations in these weights, provided this hierarchy is maintained. Ablation studies verify the contribution of each component as follows:
>
> **No Format:** The model struggles to adhere to strict syntax rules (e.g., XML tags), leading to a drastic drop in the execution rate
>
> **No Tool-Use:** The policy becomes conservative, favoring short, single-turn responses and underutilizing self-correction mechanisms.
>
> **No Visual:** Code is executable but lacks semantic alignment, producing charts with incorrect content or blank layouts.
>
>
>
> ---
>
> 1. Zheng, Ziwei, et al. "DeepEyes: Incentivizing" Thinking with Images" via Reinforcement Learning." arXiv preprint arXiv:2505.14362 (2025).
>
>
>
>
> ---
>
> Once again, we thank you for your constructive comments, which have helped us significantly improve the paper. All newly added content has been incorporated into the updated revision and highlighted in RED.

---

> ### Author Response · Authors · 2025-11-26
> **Short Summary of Discussion**
>
> Dear Reviewer 9ChV, thank you for your helpful suggestions to improve completeness. As the discussion phase is ending and our detailed response has been available for about a week, we kindly invite you to review the updates we made in response to your comments :) .
>
> **(1) Training Cost & Inference Overhead:** We clarified that the model operates in **tool-free single-pass mode** during inference, ensuring ~0 additional latency. The total training budget is ~42 hours on 8×A100 GPUs.
>
> **(2) Generalization to Broader Tasks:** We extended evaluations to **MathVerse/MathVista/LogicVista**. The results demonstrate that RRVF maintains general reasoning capabilities (49.5 vs. 49.2 on MathVerse), while SFT suffers from catastrophic forgetting (45.7, dropping ~3.5 points).
>
> **(3) Reward Design:** We corrected the reward weight typo and provided **ablation studies** on each reward component. The results confirm the principled hierarchical design: Format ensures executability, Visual measures semantic correctness, and Tool-Use encourages exploration.
>
>
> **We sincerely hope these address your concerns, could you kindly confirm whether we’ve adequately resolved them, or if further clarifications would be helpful?** Thank you for your time and invaluable insights.

---

### Author Response · Authors · 2025-11-20
**Clarification on Inference Efficiency (Crucial Misunderstanding)**

A common concern raised by Reviewer 9ChV and Reviewer WLZB concerns the inference latency and computational overhead due to tool invocation. However, it is respectfully clarified that **the tool-use and iterative refinement are exclusively employed during the training phase**. As explicitly stated in Section 4.2.2 ("Inference Setups"), the trained model operates in a **direct, single-turn inference mode** without accessing external tools or iterative loops during testing. Consequently, the inference latency is identical to that of standard MLLMs (e.g., Qwen2.5-VL), and no additional overhead is introduced at test-time.

In addition, iterative tool calls and visual-difference feedback at test-time would provide extra information beyond the single input image. This would effectively give the model additional external hints that baseline models do not receive, and therefore compromise the fairness of the comparison. For this reason, all reported results are obtained in a pure single-pass setting without tools, so that the evaluation reflects only the capabilities internalized by the model during RRVF training.

---

### Author Response · Authors · 2025-12-01
**Updated Manuscript**

We thank all reviewers for their time and constructive feedback. In particular, we appreciate the specific acknowledgement from reviewers:

*   **All reviewers** for recognizing the **novelty and impact** of the RRVF framework.
*   **9ChV and Ztqe** for highlighting that the paper is **well-written** and easy to follow, and for acknowledging the **significant performance improvements** over SFT and strong open-source baselines.
*   **WLZB and Ztqe** for appreciating the **clear and reasonable motivation** behind the "Reasoning-Rendering-Visual-Feedback(RRVF)" design.


Based on the valuable feedback, we have addressed all concerns by conducting additional experiments and updating our manuscript.  **All latest revisions are now highlighted in BLUE** for clarity, and the main changes include:

* Clarification on Inference Efficiency (addressed to 9ChV, WLZB): To resolve the misunderstanding about inference cost, we have strengthened the explanation in **Section 4.2.2** to explicitly highlight that the model operates in tool-free, single-pass mode during inference.

* New Generalization Experiments (addressed to 9ChV, Ztqe): We added zero-shot evaluations on broader reasoning benchmarks (MathVerse, MathVista, LogicVista). The results demonstrate that RRVF preserves general capabilities, strictly outperforming SFT (see **Section 4.5.2**).

* Comprehensive Ablation Studies (addressed to 9ChV, WLZB): We included detailed ablations on reasoning turns (**Section 4.5.3**) and reward components (**Appendix E**), while also verifying robustness across backbones (**Appendix D**).

* Training Cost Analysis (addressed to 9ChV, WLZB): We added a detailed breakdown of the training budget in **Section 4.4**.


Again, we thank all Reviewers and Area Chairs for their efforts and time.

Best regards,

The Authors

---

### Author Response · Authors · 2025-12-02
**Comprehensive response to AC [1/3]**

Dear Area Chairs,

We sincerely appreciate your time and effort under these unprecedented circumstances. To assist your assessment, we have prepared a comprehensive summary of our submission.



## 1. Paper Summary
This paper introduces **RRVF** (Reasoning, Rendering, and Visual Feedback), a novel framework that enables Multimodal Large Language Models (MLLMs) to learn complex visual reasoning (specifically image-to-code generation) using only raw images, without paired image-text annotations. The core innovation relies on the **"Asymmetry of Verification"**: it is easier to verify a rendered output against a source image than to reason and generate the code from scratch. We utilize this principle to provide reward signals for Reinforcement Learning (using the GRPO algorithm). Extensive experiments on chart-to-code and web-to-code tasks demonstrate that RRVF significantly outperforms supervised fine-tuning and competitive baselines, while maintaining general reasoning capabilities on broader benchmarks.


## 2. The Strengths Summarized by Reviewers

All reviewers reached a consensus on the novelty, clarity, and potential impact of our work.

*   **Novelty & Impact:** Reviewer **9ChV** highlights that "the research direction is impactful," and Reviewer **WLZB** finds the rendering-verifying process "an interesting and reasonable idea."
*   **Performance:** Reviewer **Ztqe** notes that "experiments... demonstrate that the proposed method brings significant improvements," and Reviewer **WLZB** confirms the framework "brings more significant advantages than supervised training (SFT)."
*   **Clarity:** Reviewer **Ztqe** states the paper is "overall well-written and easy to follow," and Reviewer **9ChV** agrees it is "well-written."

---

> ### Author Response · Authors · 2025-12-02
> **Comprehensive response to AC [2/3]**
>
> ## 3. Summary of the Reviewers' Concerns
>
> > Reviewer Ztqe (Initial Score: 8)
>
> | Concerns | Response|
> |---|---|
> | **W1**: Generalization(e.g., MathVista, MathVerse, LogicVista) | **New Experiment**: We add zero-shot tests on these benchmarks. RRVF matches or slightly exceeds the base model, while SFT degrades. **Revision**: Sec 4.5.2. |
>
> > Reviewer 9ChV (Initial Score: 4)
>
> | Concerns | Response |
> |---|---|
> | **W1&Q2**: Inference efficiency at test time | **Clarification:** We address a core **misunderstanding**. Inference is tool-free and single-pass, with latency identical to the base MLLM. Tool use is exclusive to the training phase.  |
> | **Q1**:Training budget | **Revision:** We report the full budget in Sec 4.4: 42 hours on 8xA100 GPUs. |
> | **W2&Q3**: Generalization | **New Experiment:** As mentioned above, we demonstrate RRVF preserves general reasoning. **Revision:** Sec 4.5.2 |
> | **W3**: Reward stability analysis | **New Experiment:** We add component-wise ablations for the reward function. Results confirm the necessity of Format, Visual, and Tool-use rewards. **Revision:** Appendix E. |
>
>
> > Reviewer WLZB (Initial Score: 4)
>
> | Concerns | Response |
> |---|---|
> | **W1**: Training & inference time | **Clarification:** We address a core **misunderstanding**. Inference is tool-free (single-pass) with **no overhead**. Training takes 42h on 8×A100.  |
> | **W2&W3**: Applicability | **New Experiments:**  RRVF demonstrates strong zero-shot OOD transfer to Plot2Code, and preserves capabilities on MathVerse/Vista. **Revision:** Sec 4.5. |
> | **W4**: Baseline coverage | **New Experiment:** Validated on Qwen2-VL-7B, proving backbone-agnostic effectiveness. **Revision:** Appendix D. |
> | **W5**: Ablations on rounds & components | **New Experiment:** Ablation on T_max verifies the effectiveness of multi-turn self-correction. Component ablations confirm hybrid rewards are essential. **Revision:** Sec 4.5.3/Appendix E. |
> | **Q2**: Steps & Overfitting | **Clarification:** Multi-turn loops provide dense supervision. Strong OOD performance confirms no overfitting. **Revision:** Sec 4.4/4.5. |

---

> ### Author Response · Authors · 2025-12-02
> **Comprehensive response to AC [3/3]**
>
> ## 4.  Final Claim
>
> We appreciate the opportunity to present this concise summary and we reaffirm that we have strictly adhered to ICLR’s Code of Ethics and the double blind policy throughout this process.
>
> The rebuttal process has been invaluable for strengthening our paper.  We clarify the key misunderstanding on inference efficiency and confirm that our method adds no overhead. We demonstrate robust generalization with extensive new experiments. We believe we have comprehensively addressed all reviewer concerns. However, due to the system-wide events, no reviewers replied or raised new questions before the comment function closed.
>
> We hope that our thoroughness and the resulting improvements merit your positive assessment. All new content is marked in **BLUE** in the revised manuscript to facilitate your review.

---

### Meta-Review · Area_Chair_v2Wx · 2025-12-12

**Summary:**

This paper studies the image-to-code problem and proposes a multi-round rendering and verification framework that eliminates the need for manually labeled text instructions. The method achieves substantial improvements over baseline models in both chat-to-code and web-to-code settings, and reviewers generally agree that the read–verify idea is interesting and reasonable. However, all reviewers raise critical concerns about the model’s generalization to broader visual reasoning tasks. During the discussion phase, the authors provided additional results on several out-of-domain benchmarks, but the added experiments remain limited in scope and do not show meaningful improvements over baselines. The work would benefit from more comprehensive evaluation on general visual tasks—such as VQA, object detection, grounding, and other reasoning-heavy benchmarks—to fully validate the proposed approach.

**Reviewer Concerns:**

Across reviewers, several major issues were identified. The concern regarding computational cost has been largely addressed in the rebuttal and discussion. However, other key concerns remain outstanding. In particular, all reviewers note limited evaluation beyond the two primary tasks, and the additional experiments provided during the discussion—while helpful—are still narrow in scope and do not demonstrate clear improvements over baseline models. The concerns about generalization to broader visual or reasoning tasks therefore, remain largely unresolved. Additionally, the issues of missing ablation studies and insufficient hyperparameter sensitivity analysis also persist, as the authors did not provide substantial new evidence or analyses.

**Reviewer Scores:**

Reviewer 9ChV gave an initial score of 4, highlighting concerns about computational cost, generalization beyond chat-to-code and web-to-code, and missing ablations. Although the computation-cost concern has been addressed, the remaining issues persist, and the reviewer is likely to keep the score at 4. Reviewer WLZB also gave a score of 4, citing concerns about computation cost, practical application scope, generalization ability, limited baselines, and missing ablations. While the authors added experiments on MathVerse, MathVista, and LogicVista, these results have a limited scope and do not demonstrate notable improvements over baselines; the added baseline is insufficient as well. This reviewer is also expected to keep the score at 4. Reviewer Ztqe initially scored the paper an 8, but their major concern—generalization to other reasoning-related benchmarks—remains insufficiently addressed, as the added experiments are limited in scope and fail to show significant improvement. This reviewer may lower the score to 6.

---

### Decision · Program_Chairs · 2026-01-26

Reject